# Ectopic Expression of *Jatropha curcas TREHALOSE-6-PHOSPHATE PHOSPHATASE J* Causes Late-Flowering and Heterostylous Phenotypes in *Arabidopsis* but not in *Jatropha*

**DOI:** 10.3390/ijms20092165

**Published:** 2019-05-01

**Authors:** Mei-Li Zhao, Jun Ni, Mao-Sheng Chen, Zeng-Fu Xu

**Affiliations:** 1CAS Key Laboratory of Tropical Plant Resources and Sustainable Use, Xishuangbanna Tropical Botanical Garden, The Innovative Academy of Seed Design, Chinese Academy of Sciences, Menglun, Mengla, Yunnan 666303, China; zhaomeili@xtbg.ac.cn (M.-L.Z.); nijun@ipp.ac.cn (J.N.); 2College of Life Sciences, University of Chinese Academy of Sciences, Beijing 100049, China

**Keywords:** trehalose-6-phosphate phosphatase, physic nut, heterostylous, late flowering

## Abstract

Trehalose-6-phosphate (T6P) phosphatase (TPP), a dephosphorylating enzyme, catalyzes the dephosphorylation of T6P, generating trehalose. In *Jatropha*, we found six members of the *TPP* family. Five of them *JcTPPA*, *JcTPPC*, *JcTPPD*, *JcTPPG*, and *JcTPPJ* are highly expressed in female flowers or male flowers, or both, suggesting that members of the *JcTPP* family may participate in flower development in *Jatropha*. The wide expression of *JcTPPJ* gene in various organs implied its versatile roles and thus was chosen for unraveling its biological functions during developmental process. We constructed an overexpression vector of *JcTPPJ* cDNA driven by the cauliflower mosaic virus (CaMV) 35S promoter for genetic transformation. Compared with control *Arabidopsis* plants, *35S:JcTPPJ* transgenic *Arabidopsis* plants presented greater sucrose contents in their inflorescences and displayed late-flowering and heterostylous phenotypes. Exogenous application of sucrose to the inflorescence buds of wild-type *Arabidopsis* repressed the development of the perianth and filaments, with a phenocopy of the *35S:JcTPPJ* transgenic *Arabidopsis*. These results suggested that the significantly increased sucrose level in the inflorescence caused (or induced) by *JcTTPJ* overexpression, was responsible for the formation of heterostylous flower phenotype. However, *35S:JcTPPJ* transgenic *Jatropha* displayed no obvious phenotypic changes, implying that *JcTPPJ* alone may not be sufficient for regulating flower development in *Jatropha*. Our results are helpful for understanding the function of *TPPs*, which may regulate flower organ development by manipulating the sucrose status in plants.

## 1. Introduction

Trehalose, a stable non-reducing α-d-glucopyranosyl α-d-glucopyranoside, is present in all organisms except vertebrates [1,2]. Trehalose has various biological functions that range from serving as an energy source to serving as a protective/signaling sugar in a variety of organisms [1,3,4,5]. Trehalose biosynthesis involves in two consecutive enzymatic reactions. First, trehalose-6-phosphate (T6P) synthase (TPS) catalyzes a reaction between both uridine diphosphate glucose (UDP-Glu) and glucose-6-phosphate, which produces the intermediate product T6P. Second, T6P phosphatase (TPP) catalyzes the dephosphorylation of T6P to generate trehalose [1]. Trehalase (TRE) then catalyzes the hydrolysis of trehalose into two units of glucose [1,6]. *Arabidopsis thaliana* has eleven *TPS* genes (*AtTPS1-11*, class I and II trehalose biosynthesis genes), ten *TPP* genes (*TPPA-J*, class III trehalose biosynthesis genes) and one *TRE* gene [7,8]. The *TPS* and *TPP* gene families are present in other flowering plants [9,10,11,12].

The presence of trehalose in a wide variety of organisms suggests a pivotal and ancient role concerning trehalose metabolism in the plant kingdom. Blázquez et al. [13] isolated the *AtTPS1* gene from *Arabidopsis*, which encodes a T6P synthase. As an endogenous substance, trehalose, along with TRE, regulates carbohydrate allocation in *Arabidopsis* [6]. In plants, disruption of trehalose metabolism can cause pleiotropic effects, including effects on carbon utilization and metabolism [14,15], cell division and cell wall synthesis [16], embryo and seedling development [17,18], leaf and inflorescence morphogenesis [19,20], the transition from vegetative growth to flowering [21], the photosynthesis process [22], and abiotic stress [23,24,25,26,27,28]. In *Arabidopsis*, transgenic plants overexpressing bacteria *trehalose 6-phosphate synthase* (*otsA*) display small, dark green leaves and early-flowering phenotypes [14]; however, transgenic *Arabidopsis* plants overexpressing bacteria *trehalose 6-phosphate phosphatase* (*otsB*) display relatively larger, lighter green leaves and late-flowering phenotypes [14]. In maize, *RAMOSA3* (*RA3*) encodes a TPP enzyme that controls maize inflorescence architecture by mediating the axillary meristems via modulation of trehalose and T6P levels [19]. Owing to their enlarged abnormal axillary meristems, *ra3* mutants display long branches or increased numbers of floral meristems [19].

Because of its non-reducing feature and the stability of the α,α-1,1 linkage, trehalose might be involved in the response to stress [29]. In rice, overexpression of *OsTPPs* improves grain yield under well-watered growth conditions and mild drought conditions, and significantly increases grain yield under severe drought conditions at the highly susceptible flowering stage [27]. Overexpression of *OsTPP7* enhances anaerobic germination by promoting starch mobilization to drive the growth kinetics of both germinating embryos and elongating coleoptiles [30]. Overexpression of *AtTPPD* results in more tolerance to high-salinity stress, whereas loss-of-function in *AtTPPD* causes hypersensitivity to salinity stress [8]. *AtTPPG* also involves in abscisic acid (ABA) related stomatal closure [12].

Sugar sensing and signaling may be involved in various aspects of growth and development in higher plants [31,32]. Wang et al. [33] reported that transgenic *Arabidopsis* lines overexpressing maize *Sucrose non-fermenting-1 (SNF1)-related protein kinase 1* (*ZmSnRK1.1*), *ZmSnRK1.2*, or *ZmSnRK1.3* accumulate high sucrose content in rosette leaves and mature seeds, and display a shortened flowering time in *ZmSnRK1.1* lines and a prolonged flowering time in *ZmSnRK1.2* and *ZmSnRK1.3* lines, respectively. In *Arabidopsis*, *SnRK1* positively regulates *FUSCA3* (*FUS3*) stability and *INDETERMINATE DOMAIN 8* (*IDD8*) phosphorylation to mediate flowering time; *FUS3* encodes a B3-domain transcription factor regulating organ (e.g., embryo, cotyledon, silique, and floral organ) formation and developmental phase transition, and *IDD8* regulates sucrose synthesis (*SUC*) gene expression to affect photoperiodic flowering [34,35,36,37]. In cucumber (*Cucumis sativus*), downregulation of sucrose transporter (*CsSUT1*) can decrease sucrose content in male flowers, and perturb male flower development generating smaller flower [38].

*Jatropha curcas* L., belonging to the Euphorbiaceae family, is a perennial woody plant and widely distributes in tropical and subtropical regions. *Jatropha* has a dichasial cyme inflorescence bearing 0–10 female and 25–215 male flowers [39]. Owing to high oil content in seeds, *Jatropha* is considered as a promising renewable energy plant [40,41,42,43,44,45,46]. In this study, to validate its function, we isolated a *JcTPPJ* gene from *Jatropha* and overexpressed it in *Arabidopsis* and *Jatropha* for its functional validation. The transgenic *Arabidopsis* plants displayed heterostylous flowers and late-flowering phenotypes; however, the transgenic *Jatropha* plants did not display obvious phenotype change.

## 2. Results

### 2.1. JcTPPJ is Highly Expressed in Male Flowers

A total of six genes *JcTPPA*, *JcTPPC*, *JcTPPD*, *JcTPP1*, *JcTPPG*, and *JcTPPJ* that are homologous to *Arabidopsis trehalose-6-phosphate phosphatase* (*AtTPP*) genes were identified from *Jatropha*. JcTPPs contain a HAD-like (haloacid dehalogenase-like hydrolase) domain similar to that in *Escherichia coli* trehalose 6-phosphate phosphatase (otsB) [47,48]. Phylogenetic analysis revealed that JcTPP family members could be divided into two groups: clade I included AtTPPB, AtTPPC, AtTPPE, AtTPPH, AtTPPI, AtTPPJ, JcTPPC, JcTPPD, and JcTPPJ, and clade II included AtTPPA, AtTPPD, AtTPPF, AtTPPG, JcTPPA, JcTPP1, and JcTPPG (Figure 1).

The transcriptional level of the *JcTPPs* was further investigated in different tissues, including roots, stems, young leaves, mature leaves, stem apices, female flowers, male flowers, and fruits using quantitative reverse transcriptase-polymerase chain reaction (qRT-PCR). The *JcTPPJ* gene was broadly expressed in all these tissues, and particularly high expressed in roots, stems, and male flower tissues (Figure 2). *JcTPPA*, *JcTPPC*, *JcTPPD*, and *JcTPPG* were highly expressed in the flowers and *JcTPP1* in the inflorescence (Appendix A). These results suggested that members of the *JcTPP* family might be involved in the process of flower development in *Jatropha*. In *Arabidopsis* and rice, *AtTPPC*, *AtTPPD*, *AtTPPG*, and rice *TPP1* were shown to regulate flower development, and *35S:AtTPPB*, *35S:AtTPPC* and *35S:AtTPPI* transgenic plants displayed short-perianth phenotypes [8,12,27]. However, the expression level of the *JcTPP* members, except *JcTPPG*, was very low in various tissues of *Jatropha* (Figure 2 and Appendix A). Similarly, the expression of *Arabidopsis AtTPPC* and rice *OsTPP1* is extremely low in the most tissues [49,50]. In grapevine, *VvTPPF* cannot be detected in any organs and *VvTPPC* and *VvTPPD* are detected having a very low level in leaves and stems [51]. The wide expression of *JcTPPJ* gene indicates it may involve in the regulation of multiple developmental processes. Thus, we selected the *JcTPPJ* gene to further investigate its biological function.

### 2.2. Overexpression of JcTPPJ in Arabidopsis Delayed Flowering

To investigate the functionality of *JcTPPJ*, transgenic *Arabidopsis* and *Jatropha* lines expressing *JcTPPJ* under the control of the cauliflower mosaic virus (CaMV) 35S promoter were obtained. Transgenic *Arabidopsis* plants were selected in the presence of kanamycin, and the expression level of *JcTPPJ* was validated via qRT-PCR (Figure 3). In total, twelve independent *35S:JcTPPJ* T0 transgenic lines were obtained. Compared with the control plants, the *35S:JcTPPJ* transgenic plants displayed a late-flowering phenotype under inductive long-day conditions (Figure 4A). We then selected three homozygous transgenic lines of the T3 generation for further phenotypic examination. The flowering of *35S:JcTPPJ* transgenic plants was delayed for two to three days (Figure 4B), although these plants had more rosette leaves than the control plants did under long-day conditions (Figure 4C). These results showed that *JcTPPJ* participates in the regulation of flowering in *Arabidopsis*.

### 2.3. Overexpression of JcTPPJ in Arabidopsis Resulted in Delayed Perianth and Stamen Filaments Development

The *35S:JcTPPJ* transgenic plants displayed a heterostylous phenotype (Figure 5A). The perianth and stamen filaments were shorter than those of the controls, but the length of the stigmas was not obviously different (Figure 5B,C). The results showed that the development of perianth and stamen filaments in *Arabidopsis* was suppressed by *JcTPPJ*. We also treated the inflorescence buds of wild-type (WT) *Arabidopsis* with 50 mM sucrose and trehalose solutions, respectively. Both treatments suppressed the development of the perianth and stamen filaments, and the sucrose treatment caused a more obvious phenotype (Figure 6). In addition, we measured the sucrose content in the inflorescences of the transgenic and control plants. The sucrose content in the inflorescences was greater in the transgenic plants than in the control plants, implying that overexpression of *JcTPPJ* can promote the accumulation of soluble sugars in *Arabidopsis* (Figure 7A). This observation suggested that the inhibition of perianth and stamen filaments development in the transgenic *Arabidopsis* plants was caused by the high soluble sugar content, which resulted from the overexpression of *JcTPPJ*.

### 2.4. Overexpression of JcTPPJ Did Not Affect Flower Development in Jatropha

We also validated the functionality of *JcTPPJ* in *Jatropha*. Nine independent *35S:JcTPPJ* transgenic T0-generation lines were obtained and confirmed via qRT-PCR (Figure 8). Three positive lines were selected for phenotypic observation; however, the transgenic plants did not display visible phenotypic changes (Figure 9), indicating that *JcTPPJ* alone may be insufficient to affect flower development in *Jatropha* or that the functionality of *JcTPPJ* differs between *Jatropha* and *Arabidopsis* systems. The sucrose content in the inflorescences between WT and transgenic *Jatropha* had no significant difference, implying that overexpression of *JcTPPJ* may be insufficient for promoting the accumulation of soluble sugars in *Jatropha* (Figure 7B).

## 3. Discussion

The stigmas and anthers of heterostylous species develop separately (herkogamy) for preventing self-pollination and promoting outcrossing, which aids in environmental adaption and species diversity [52,53,54,55,56]. The morphology of the gynoecia atop the anthers is considered to be long-styled [52,57], and that of the stigmas below the anthers is short-styled. Transgenic *Arabidopsis* plants overexpressing *JcTPPJ* display the long-styled morphology (Figure 5), which suggests that *JcTPPJ* or pathways related to *JcTPPJ* might participate in the formation of heterostyly in heterostylous species.

Overexpression of *TPP* can promote growth and varying severity of morphological abnormalities in several species [26,27,30,50]. The perianth of transgenic *Arabidopsis* plants overexpressing *AtTPPB*, *AtTPPC*, and *AtTPPI* is relatively short, preventing the protection and covering of the pistil and stamen by the perianth [12]. Overexpressing *JcTPPJ* in *Arabidopsis* results in phenotypes similar to those of *AtTPPB*, *AtTPPC*, and *AtTPPI* transgenic plants; however, the filaments of the stamen are relatively short in *35S:JcTPPJ* transgenic *Arabidopsis*, which implies that *JcTPPJ* plays a stronger role in preventing the development of male floral organs than does *AtTPPB*, *AtTPPC*, and *AtTPPI* as assessed by their overexpression in transgenic plants (Figure 5). Moreover, Vandesteene et al. [12] reported that *AtTPPJ* is highly expressed in expanding leaves, leaf primordia, hypocotyl–root junction, and stamen, which is similar to the expression pattern of *JcTPPJ* in *Jatropha*, supporting that *JcTPPJ* and *AtTPPB*, *AtTPPC*, and *AtTPPI* share the same functionality. Trehalose-6-phosphate phosphatase (TPP) catalyzes dephosphorylation of T6P to generate trehalose [1]. Nuccio et al. [27] reported that overexpression of rice *TPP1* driven by the *OsMads6* promoter in maize increases the sucrose content in ear spikelets. Kretzschmar et al. [30] demonstrated that *OsTPP7* activity can enhance sucrose availability. In *Arabiodpsis*, the sucrose content is slightly lower in *tppj* mutants than in WT plants [12]. Overexpression of both *E. coil otsB* and *AtTPPs* in *Arabidopsis* results in a late-flowering phenotype, respectively [12,14]. Ohto, et al. [31] reported that sucrose may affect the transition from vegetative growth to reproductive growth by activating or inhibiting the expression of flowering genes, which depend on sugar concentration, developmental period, and genetic background of plants. *Arabidopsis* plants overexpressing *SNF1 KINASE HOMOLOG 10* (*AKIN10*) encoding a sucrose non-fermenting-1 (SNF1) related protein kinase, displayed phase transition, and organ development defects by delaying the degradation of *FUS3* [36]. Overexpression of *JcTPPJ* in *Arabidopsis* also resulted in increased sucrose contents, which may cause the development of short perianth and stamen filaments and a delay in flowering (Figure 4 and Figure 5). However, overexpression of *JcTPPJ* in *Jatropha* had no obvious phenotypic changes (Figure 9), which may be because of an insufficient increase in sucrose content in transgenic plants. And it is also possible that *JcTPPJ* may need to interact with other *JcTPP* members or other genes to regulate floral development in *Jatropha*. Five of six members of the *JcTPP* family were present in *Jatropha*, and five of them *JcTPPA*, *JcTPPC*, *JcTPPD*, *JcTPPG*, and *JcTPPJ* are highly expressed in female flowers or male flowers, or both, suggesting that they may participate in the development of flowers in *Jatropha*. This supposition was supported partly by the *JcTPPJ* transgenic *Arabidopsis* plants displaying heterostyly and late-flowering phenotypes (Figure 4 and Figure 5).

## 4. Materials and Methods

### 4.1. Plant Materials and Growth Conditions

The cultivar ‘Flowery’ of *Jatropha* and the Columbia ecotype of *Arabidopsis thaliana* were used throughout the experiment. *Jatropha* seeds were planted in pots that contain peat-based soil and then were transferred in a greenhouse. The plants were incubated at 25 ± 2 °C under long-day conditions (14 h light/10 h dark) in which lighting is provided by cool-white fluorescent bulbs. Mature *Jatropha* plants were transplanted into an experimental field at the Xishuangbanna Tropical Botanical Garden (21°54′ N, 101°46′ E), Chinese Academy of Sciences [58]. The photosynthetically active radiation reached 1850 μmol m^−2^ s^−1^ in the summer and 1550 μmol m^−2^ s^−1^ in the winter [59]. The *Arabidopsis* seeds were germinated on half-strength Murashige and Skoog medium (1/2 MS). First, the seeds were vernalized for 2 days at 4 °C, and then were transferred to program ray radiation incubator (Percival, E-36L2) under a long-day (16 h light/8 h dark) photoperiod with 22 °C/20 °C day/night temperatures for seven days. The seedlings were transplanted to peat-based soil and maintained in program ray radiation incubator. We collected the roots, stems, mature leaves, young leaves, stem apices, male flowers, female flowers, and fruits of WT *Jatropha*; the young leaves of transgenic *Jatropha*; and the rosette leaves of transgenic *Arabidopsis* for qRT-PCR, all of which were immediately frozen in liquid nitrogen and then stored at −80 °C until use.

### 4.2. Phenotypic Analysis

The phenotypes of homozygous (T3) *Arabidopsis* plants and heterozygous (T0 and T1) *Jatropha* plants were analyzed. When the main inflorescence shoot had elongated past 1.0 cm, we recorded the flowering time and counted the numbers of *Arabidopsis* rosette leaves. Twenty plants were used to investigate genotypes. For the flower phenotypes, the flower buds were dissected for microscopic analysis (3D super-depth digital microscope, ZEISS, Smart zoom 5).

### 4.3. Isolation of JcTPPJ cDNA

To isolate the *JcTPPJ* gene, total RNA was extracted from mature leaves of *Jatropha* with a pBIOZOL RNA extraction reagent (Bioflux, Hangzhou, China), and the quality of RNA was measured by a NanoDrop 2000 spectrophotometer (Thermo Fisher, Wilmington, DE, USA). In accordance with the TAKARA PrimeScript™ RT Reagent Kit (TAKARA Biotechnology, Dalian, China), we used approximately 1.0 µg of total RNA for cDNA synthesis. The full-length coding sequence of *JcTPPJ* was obtained from this cDNA library by PCR with TransStart KD Plus DNA Polymerase (TRANSGEN Biotech, Beijing, China) and gene-specific primers. A combination of XC9 and XC10 primers were used (Appendix A). The PCR products were subsequently gel purified, digested with appropriate restriction enzymes, cloned into a pEASY-Blunt Zero cloning vector (TRANSGEN Biotech, Beijing, China), and sequenced. The *JcTPPJ* mRNA sequence has been deposited in GenBank under accession number MK587444.

### 4.4. Phylogenetic Analysis and Sequence Alignment

The full-length protein sequences of JcTPPA, JcTPPC, JcTPPD, JcTPP1, JcTPPG, JcTPS1, AtTPS1, and AtTPPA-J and bacteria otsB were downloaded from the NCBI GenBank (Appendix A). We then used ClustalX2 (http://www.clustal.org/clustal2/) to perform sequence alignment, and manually edited using BioEdit software (https://bioedit.software.informer.com/). MEGA (version 5.0) software (http://www.megasoftware.net/) was used to construct a phylogenetic tree based on the N-J method with 1000 bootstrap replications; the bootstrap percentages were shown on the dendrogram branch points. The sequences of JcTPS1 and AtTPS1 were used as outgroups.

### 4.5. Vector Construction and Transformation

We constructed a *35S:JcTPPJ* plant overexpression vector. The confirmed cDNA sequence was excised from the sequenced vector using appropriate restriction enzymes and then cloned into a pOCA30 binary vector with the CaMV 35S promoter. The vector was introduced into *Agrobacterium tumefaciens* strain EHA105 via the freeze–thaw method [32], and Col-0 *Arabidopsis* plants were transformed by the floral dip method [60]. To identify the transgenic lines, seeds were germinated on 1/2 MS medium supplemented with 50 mg/L kanamycin, and the survivors were genotyped to confirm. Genetic transformation of *Jatropha* was performed using *Agrobacterium*-mediated methods described as Fu, et al. [61]. Transgenic *Jatropha* plants were confirmed by genomic PCR and qRT-PCR methods.

### 4.6. Quantitative RT-PCR Analysis

To investigate the expression patterns of *JcTPPJ* and identify positive *35S:JcTPPJ* transgenic plants, qRT-PCR was performed using SYBR^®^ Premix Ex Taq™ II (TAKARA Biotechnology, Dalian, China) on a LightCycler 480II (Roche Diagnostics, Mannheim, Germany) device. The *JcGAPDH* [62] and *AtActin2* genes were used to normalize the transcript levels of specific genes of *Jatropha* and *Arabidopsis*, respectively. At least three biological replicates were used for all samples. The 2^−ΔΔ*C*t^ method was used to analyze the data described as Livak and Schmittgen [63]. The Δ*Ct* value was from difference between *Ct* values of reference gene *JcGAPDH* or *AtActin2* and each target gene; the ΔΔ*C*t was calculated by subtracting the mean Δ*Ct* of the control samples from mean Δ*Ct* of transgenic samples. Primers for qRT-PCR are listed in Appendix A.

### 4.7. Sucrose Treatment and Determination

Sucrose and trehalose were respectively dissolved in water for preparation of 50 mM working solutions containing 0.05% (*v*/*v*) Tween-20 (BBI, Shanghai, China). For sucrose or trehalose treatment experiment, 10 μL of the working solutions was directly dropped onto the flowers of four-week-old *Arabidopsis* plants, respectively. Control plants were treated with distilled water solutions that contained the same concentration of Tween-20. After one to two days, phenotypes of the treated flowers were investigated.

For the extraction of sucrose compounds, the inflorescences were harvested from four-week-old *Arabidopsis* plants and adult *Jatropha* plants. Sucrose isolation was performed using a plant sucrose content determination kit (Solarbio, Beijing, China [64]). Approximately 0.1 g samples were crushed at room temperature and ground with 0.5 mL of extract solution. After grinding, the homogenates were quickly transferred to a centrifugal tube incubating at 80 °C for 10 min. During this process, the tubes were shaken 3–5 times. After cooled to the room temperature, the tubes were subjected to centrifugation (4000× *g*, 10 min, 25 °C). The supernatants were collected and decolorized by adding 2 mg of Reagent Five at 80 °C to 1.0 mL of extract solution, incubating for 30 min. After centrifugation (4000× *g*, 10 min, 25 °C), the supernatants were collected for sucrose determination using a NanoDrop 2000 spectrophotometer (Thermo Fisher, Wilmington, DE, USA).

### 4.8. Statistical Analyses

Results are expressed as the means ± SE of data. The relative expression level of genes was obtained from at least three biological replicates for each sample as above described. Data for sucrose determination experiment were obtained from at least three independent experiments. Data for days and rosette leaves were obtained from 16 independent plants. One-way ANOVA was used to compare the differences between means of samples and to determine the statistical significance (* *p* < 0.05, ** *p* < 0.01).

## 5. Conclusions

The *35S:JcTPPJ* transgenic *Arabidopsis* plants having high sucrose contents displayed late-flowering and heterostyly phenotypes. Exogenous application of sucrose to *Arabidopsis* inflorescence buds repressed perianth and filaments development. These results indicate that the heterostyly of *35S:JcTPPJ* transgenic plants is caused by the high sucrose content in the inflorescence and sucrose accumulation is promoted by the overexpression of *JcTPPJ* in *Arabidopsis*. However, compared with that of WT *Jatropha*, the phenotype of *35S:JcTPPJ* transgenic *Jatropha* had no obvious phenotypic changes, indicating that *JcTPPJ* alone is insufficient to alter the phenotypes of flowers in *Jatropha*. Nevertheless, the expression patterns of *JcTPP* family members suggest that *JcTPPJ* together with the other *JcTPPs* may play important roles in regulating flower development in *Jatropha*. Further work, such as knockout of *JcTPPJ* by CRISPR-Cas9 system [42] or knockdown of *JcTPPJ* by RNA interference (RNAi) method [41], will be needed to demonstrate the involvement of *JcTTPJ* in *Jatropha* flower development.

## Figures and Tables

**Figure 1 ijms-20-02165-f001:**
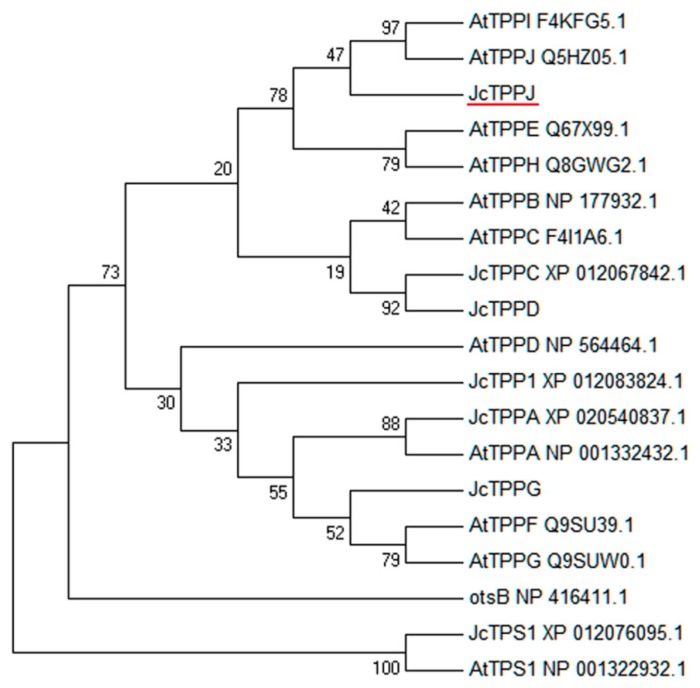
Phylogenetic analysis of TPP family members in *Jatropha*, *Arabidopsis*, and bacterium, and the amino acid sequences of JcTPS1 and AtTPS1 were used as outgroups. The tree was constructed by the neighbor-joining (N-J) method. The GenBank accession numbers are listed in Appendix A.

**Figure 2 ijms-20-02165-f002:**
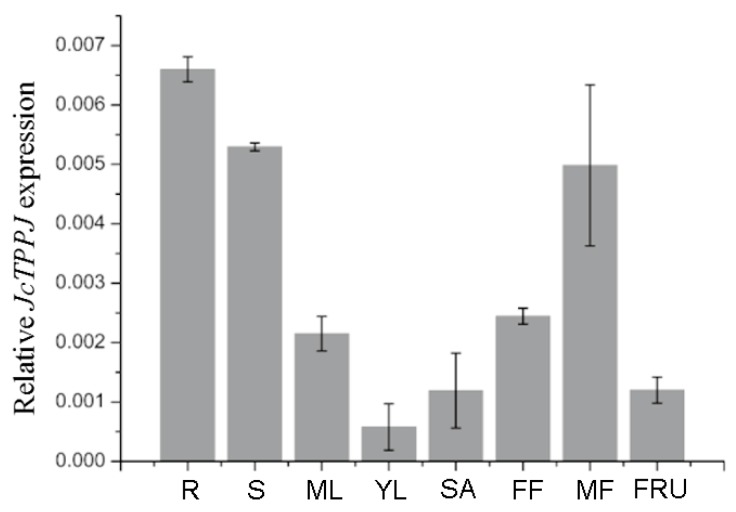
The transcriptional level of *JcTPPJ* in various organs of adult *Jatropha* plant. The qRT-PCR results were obtained from three independent biological replicates per sample. The levels of detected amplification were normalized via the amplified products of the *JcGAPDH* genes as a reference. The error bars represent the standard errors. R, roots; S, stems; ML, mature leaves; YL, young leaves; SA, stem apices; FF, female flowers; MF, male flowers; FRU, fruits.

**Figure 3 ijms-20-02165-f003:**
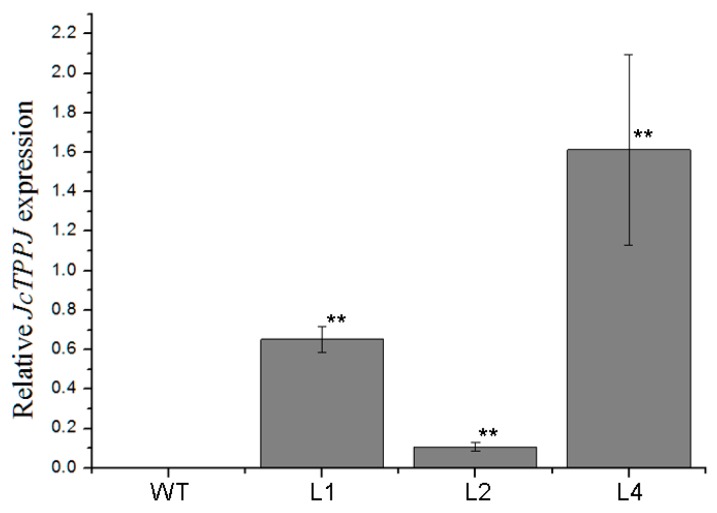
Relative expression levels of *JcTPPJ* in wild-type (WT) and different transgenic *Arabidopsis* lines (L1, L2, and L4). The qRT-PCR results were obtained from three independent biological replicates per sample. The amplification levels were normalized using the amplified products of the *AtActin2* gene as a reference. ** indicates a significant difference at the *p* < 0.01 level determined by one-way ANOVA. The error bars represent the standard errors (*n* = 3).

**Figure 4 ijms-20-02165-f004:**
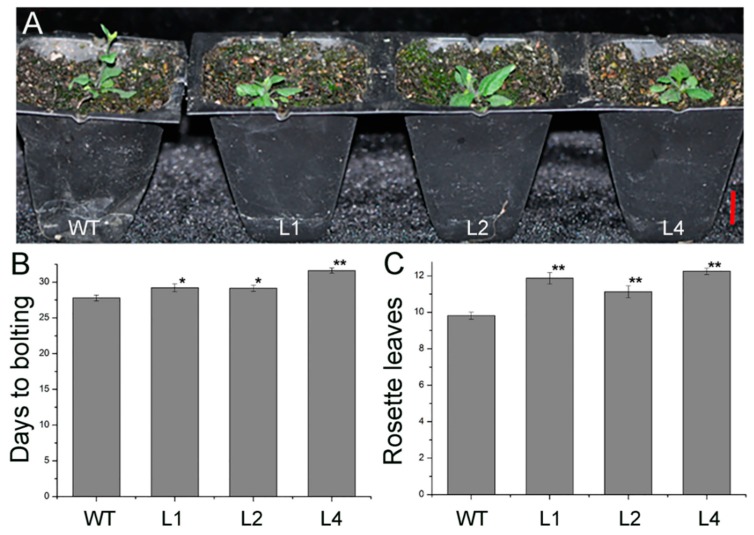
Ectopic expression of *JcTPPJ* causes late flowering in transgenic *Arabidopsis*. (**A**) Wild-type (WT) and transgenic *Arabidopsis* lines (L1, L2, and L4) grown under long-day conditions (16 h light/8 h dark) after vernalization, the bar represent 1.0 cm. Days (**B**) and rosette leaves (**C**) after the main inflorescence shoot had elongated past 1 cm in *35S:JcTPPJ* transgenic *Arabidopsis* lines grown under long-day conditions. * indicates a significant difference at the *p* < 0.05; ** indicates a significant difference at the *p* < 0.01 level determined by one-way ANOVA. The error bars represent the standard errors (*n* = 16).

**Figure 5 ijms-20-02165-f005:**
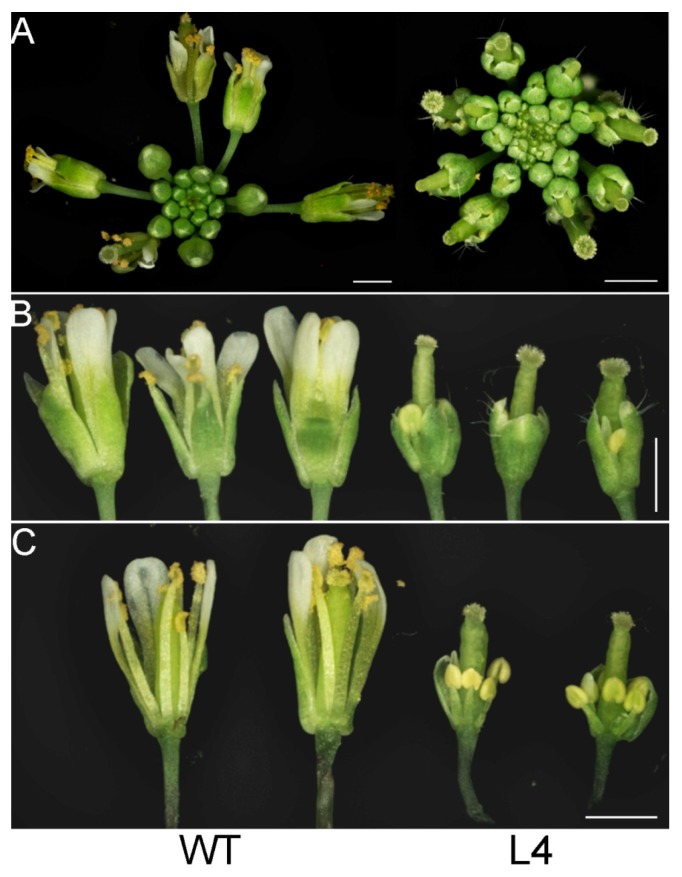
Ectopic expression of *JcTPPJ* in *Arabidopsis* affects floral organ development. Inflorescence (**A**), flowers (**B**), and dissected flowers (**C**) of wild-type (WT) and transgenic *Arabidopsis* line L4. The bars represent 1.0 mm.

**Figure 6 ijms-20-02165-f006:**
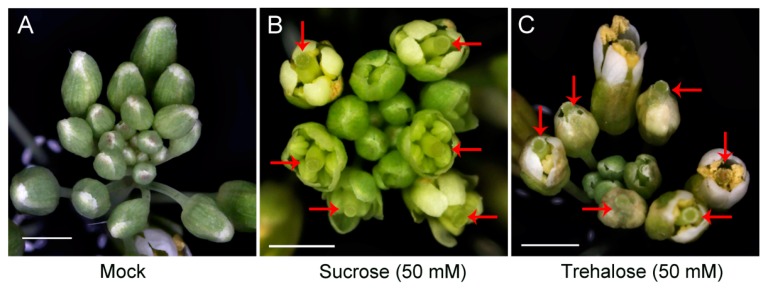
Applications of sucrose and trehalose inhibit development of perianth and stamen filaments development in *Arabidopsis*. Inflorescences treated with mock (**A**), with 50 mM sucrose (**B**), and with 50 mM trehalose (**C**). Red arrows indicate pistils. The bars represent 1.0 mm.

**Figure 7 ijms-20-02165-f007:**
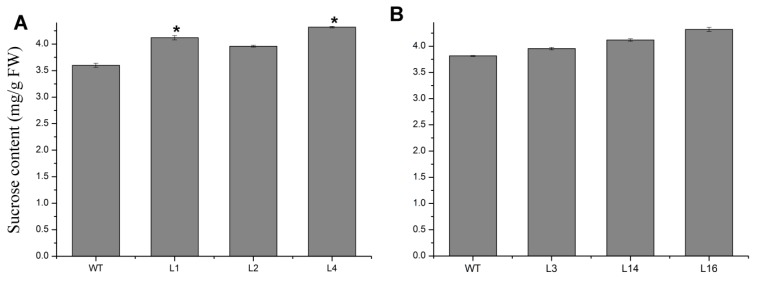
Comparison of sucrose content in the inflorescences between wild-type (WT) and transgenic *Arabidopsis* lines (L1, L2, and L4) (**A**), and between WT and transgenic *Jatropha* lines (L3, L14, and L16) (**B**). One-way ANOVA was used to determine significant differences between the transgenic and control plants. FW means fresh weight. * indicates a significant difference at the *p* < 0.05 level. The error bars represent the standard errors (*n* = 3).

**Figure 8 ijms-20-02165-f008:**
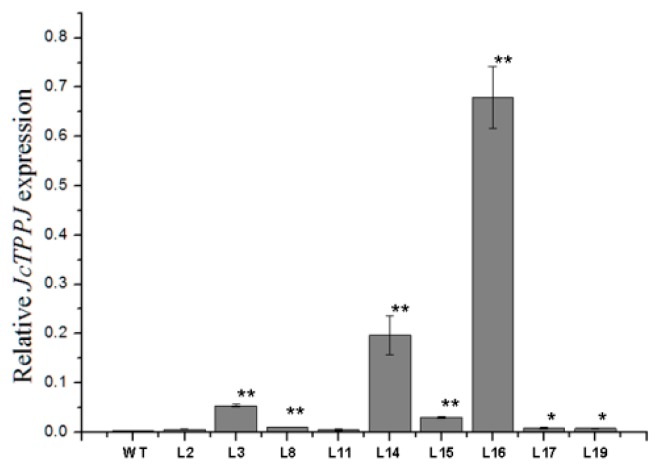
Relative expression levels of *JcTPPJ* in *35S:JcTPPJ* transgenic *Jatropha*. The qRT-PCR results were obtained from three independent biological replicates per sample from wild-type (WT) and *35S:JcTPPJ* transgenic *Jatropha* lines (L2, L3, L8, L11, L14, L15, L16, L17, and L19). The levels of detected amplification were normalized using the amplified products of the *JcGAPDH* gene as a reference. * indicates a significant difference at the *p* < 0.05 level, ** indicates a significant difference at the *p* < 0.01 level determined by one-way ANOVA. The error bars represent the standard errors (*n* = 3).

**Figure 9 ijms-20-02165-f009:**
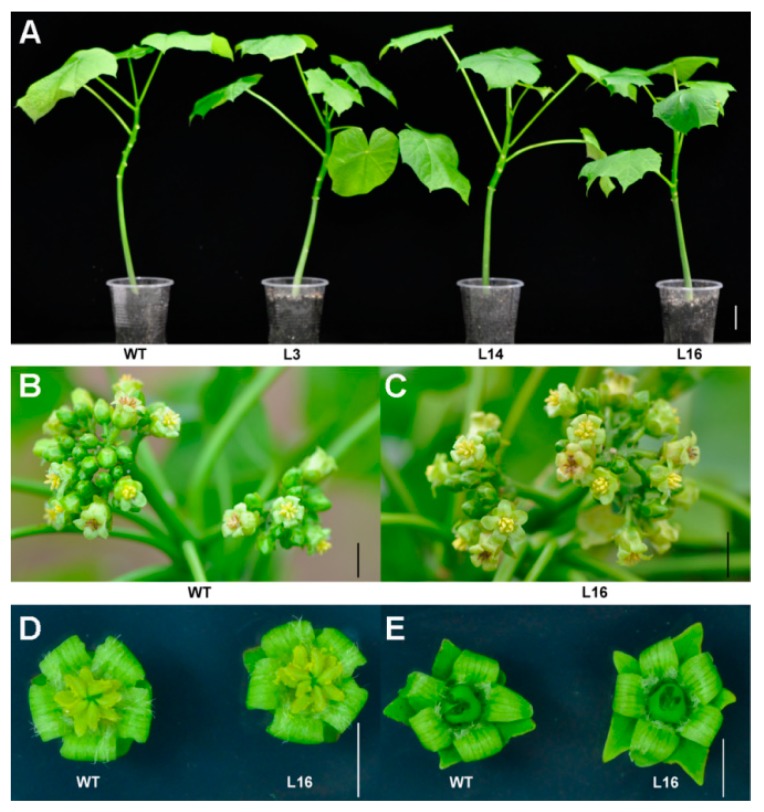
Phenotypes of *35S:JcTPPJ* transgenic *Jatropha*. The 30-day-old seedlings of wild-type (WT) and transgenic lines (L3, L14, and L16) (**A**). The inflorescences of WT (**B**) and *35S:JcTPPJ* transgenic line L16 (**C**) *Jatropha* plants in the field. Male flowers (**D**) and female flowers (**E**) of WT and transgenic line L16 plants. The bars represent 1.0 cm.

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
