# Peer review of "Ectopic Expression of Jatropha curcas TREHALOSE-6-PHOSPHATE PHOSPHATASE J Causes Late-Flowering and Heterostylous Phenotypes in Arabidopsis but not in Jatropha"

_ijms, 2019, doi:10.3390/ijms20092165_

Round 1
Reviewer 1 Report
In this work, Zhao and his colleagues try to determine possible roles of Jatropha curcas PHOSPHATASE OF TREHALOSA-6-PHOSPHATE J (JcTPPJ) in the regulation of flowering and development of flowers through ectopic expression in Jatropha and Arabidopsis thaliana. Experiments are well performed and paper properly written but authors should take care of some aspects before publication.
In my opinion, the biggest problem of the article is the statistical significance of the results shown. It is not indicated in almost any experiment of these work if the differences found are statistically significant or not. Just in figure legend 7A, it can be found some results (sucrose content in wild-type and transgenic Arabidopsis) that are significantly different. Can authors perform any statistical test in the expression and flowering time experiments which show which differences are significant and which not? Or have they performed the test and just the ones in 7A are different? If that is the case, authors should be really careful on the robustness of their conclusions. Without clear statistical differences the significance of the work would be low.
Authors should be careful with the name of genes and transgenic lines (35S:JcTPPJ), which should be in italics, all over the paper. Please, check and correct properly since some times are in italics, but not most of the times (for example in line 77, 92, etc.).
Authors must include at the end of the discussion or conclusion sections, some experimental strategies to demonstrate the involvement of JcTTPJ in Jathropa flower development.
Regarding figures, please check:
In gene relative expression pIots is necessary to explain in materials and methods or results, which is considered the expression reference level in every experiment, and indicate somewhere what indicates the Y-axis (ddCt values?). Please, clarify.
Figure 6: I cannot see a clear difference between the inflorence control (A) and the one treated with trehalose (B). Moreover, it is not stated in the legend what the red arrows mark. Can authors find a better picture for demonstrating that trehalose treatment induce changes in perianth and stamen filaments? If not, I would not say that trehalose induce detectable changes.
Figure 7: Y-axes legends are difficult to read because its small size. Can authors put in parallel the A and B panel and make them bigger?
Minor issues that author should take care are:
Indicate in line 80 that otsB gene code for the TPP of E. coli.
Line 87, change the tense of “were” for “are”.
Line 107, change was for were.
Line 109 and 111, change JCTPPJ for 35S:JcTPPJ in italics (and do it similarly in all the text).
Line 127, when filaments stated, indicate “stamen filaments”.
Line 135, change “Therefore” for “This observation suggests that…” the inhibition….
Line 190, sentence “Similarly, ….” has no sense.
Line 191 tppj should be in italics.
Line 263 add “genes” after AtActin2.
Author Response
Response to Reviewer 1
Point 1: In my opinion, the biggest problem of the article is the statistical significance of the results shown. It is not indicated in almost any experiment of these work if the differences found are statistically significant or not. Just in figure legend 7A, it can be found some results (sucrose content in wild-type and transgenic Arabidopsis) that are significantly different. Can authors perform any statistical test in the expression and flowering time experiments which show which differences are significant and which not? Or have they performed the test and just the ones in 7A are different? If that is the case, authors should be really careful on the robustness of their conclusions. Without clear statistical differences the significance of the work would be low.
Response 1: Thanks for your suggestion. We performed statistical significance analysis of the data shown in Figure 3, 4, 7 and 8, and revised the figures accordingly in the revised manuscript.
Point 2: Authors should be careful with the name of genes and transgenic lines (35S:JcTPPJ), which should be in italics, all over the paper. Please, check and correct properly since some times are in italics, but not most of the times (for example in line 77, 92, etc.).
Response 2: Thanks for your suggestion. We checked and corrected carefully all names of genes and transgenic lines throughout the revised manuscript.
Point 3: Authors must include at the end of the discussion or conclusion sections, some experimental strategies to demonstrate the involvement of JcTTPJ in Jathropa flower development.
Response 3: Thanks for your suggestion. We have included the information at the end of the conclusion section (Lines 337-339 in page 11).
Point 4: Regarding figures, please check:
In gene relative expression plots is necessary to explain in materials and methods or results, which is considered the expression reference level in every experiment, and indicate somewhere what indicates the Y-axis (ddCt values?). Please, clarify.
Response 4: We added the detailed information in materials and methods section (Lines 297-300 in page 10).
Point 5: Figure 6: I cannot see a clear difference between the inflorence control (A) and the one treated with trehalose (B). Moreover, it is not stated in the legend what the red arrows mark. Can authors find a better picture for demonstrating that trehalose treatment induce changes in perianth and stamen filaments? If not, I would not say that trehalose induce detectable changes.
Response 5: We revised the Figure 6 by using better pictures of the flowers under microscope. Description of red arrows was added in the legend (Lines 168-171 in page 6).
Point 6: Figure 7: Y-axes legends are difficult to read because its small size. Can authors put in parallel the A and B panel and make them bigger?
Response 6: We revised the Figure 7 and increased the font size.
Point 7: Minor issues that author should take care are:
Indicate in line 80 that otsB gene code for the TPP of E. coli.
Response 7: We added “Escherichia coli trehalose 6-phosphate phosphatase” describing otsB (Lines 94-95 in page 3).
Point 8: Line 87, change the tense of “were” for “are”.
Response 8:We revised the “were” as “are” (Line 102 in page 3).
Point 9: Line 107, change was for were.
Response 9: We revised the “was” as “were” (Line 128 in page 4).
Point 10: Line 109 and 111, change JcTPPJ for 35S:JcTPPJ in italics (and do it similarly in all the text).
Response 10: We revised the “JcTPPJ” as “35S:JcTPPJ” in the revised manuscript.
Point 11: Line 127, when filaments stated, indicate “stamen filaments”.
Response 11: We revised the “filaments” as “stamen filaments” in the revised manuscript.
Point 12: Line 135, change “Therefore” for “This observation suggests that…” the inhibition….
Response 12: We revised this sentence (Lines 160-161 in page 5).
Point 13: Line 190, sentence “Similarly, ….” has no sense.
Response 13: Thanks for your suggestions. We have revised this sentence (Line 221 in page 8).
Point 14: Line 191 tppj should be in italics.
Response 14: We revised “tppj” as “tppj” (Line 223 in page 9).
Point 15: Line 263 add “genes” after AtActin2.
Response 15: We added “genes” after AtActin2 (Line 295 in page 10).

Reviewer 2 Report
General Comments:
The manuscript “Ectopic expression of Jatropha curcas TREHALOSE-6-2 PHOSPHATE PHOSPHATASE J causes late-flowering 3 and heterostylous phenotypes in Arabidopsis but not 4 in Jatropha” compares the floral phenotype in Arabidopsis and Jatropha, overexpressing JcTPPJ, an enzyme involved in the generation of trehalose, as well as the effect of external sucrose application on flower development in Arabidopsis.
The manuscript is well written, and the results contribute to the understanding of TPP family members in flower development. However, both introduction and discussion lack information on the possible mechanism of sucrose in flowering time and flower organ development.
Detailed comments:
Abstract:
Please mention briefly in the abstract why you have chosen cTPPJ from the six members of the TPP family in Jatropha curcas.
P1L16 JcTPPA, JcTPPC, JcTPPD, JcTPPG, and JcTPPJ, are highly expressed in female flowers and/or male flowers –Please make clear that this is a result of the present study.
P1L 23 – please correct: …with a phenocopy
Introduction:
Please add some more information about the botany of Jatropha curcas concerning the existence of male and female flowers.
Please add some information on expression level of TPP family members. This is relevant especially considering the strikingly low relative expression level (max. 0.0065) in different tissues of Jatropha
Results
P3 L 96- please explain more in detail while authors decided to work with JcTPP. This gene seems to be more universally expressed throughout plant tissues, while other members seem to be more flower and inflorescence specific.
Fig. 4 – please indicate how many plants of the T3 generation of each line were evaluated and whether differences were significant.
The relatively low expression incrementation in L2 is not strongly reflected in phenotype – please comment on this.
Discussion:
P7L90 … demonstrated that OsTPP7 activity sucrose availability – please correct sentence.
In general, authors should discuss more in detail the function that an altered sucrose metabolism could have on flowering time and flower development.
Author Response
Response to Reviewer 2 Comments
Point 1: General Comments:
The manuscript “Ectopic expression of Jatropha curcas TREHALOSE-6-2 PHOSPHATE PHOSPHATASE J causes late-flowering 3 and heterostylous phenotypes in Arabidopsis but not 4 in Jatropha” compares the floral phenotype in Arabidopsis and Jatropha, overexpressing JcTPPJ, an enzyme involved in the generation of trehalose, as well as the effect of external sucrose application on flower development in Arabidopsis.
The manuscript is well written, and the results contribute to the understanding of TPP family members in flower development. However, both introduction and discussion lack information on the possible mechanism of sucrose in flowering time and flower organ development.
Response 1: Thanks for your suggestion. We added the information on the possible mechanism of sucrose in flowering time and flower organ development in both introduction and discussion sections (Lines 70-81 in page 2 and lines 224-230 in page 9).
Point 2: Detailed comments:
Abstract:
Please mention briefly in the abstract why you have chosen JcTPPJ from the six members of the TPP family in Jatropha curcas.
Response 2: Thanks for your suggestion. We added the relevant description in abstract section (Lines 18-20 in page 1).
Point 3: P1L16 JcTPPA, JcTPPC, JcTPPD, JcTPPG, and JcTPPJ, are highly expressed in female flowers and/or male flowers –Please make clear that this is a result of the present study.
Response 3: We have revised relevant sentence in abstract section (Lines 15-18 in page 1).
Point 4: P1L 23 – please correct: …with a phenocopy
Response 4: We have corrected it in abstract section (Line 25 in page 1).
Point 5: Introduction:
Please add some more information about the botany of Jatropha curcas concerning the existence of male and female flowers.
Response 5: We added the descriptions of the male and female flowers in Jatropha in introduction section (Lines 83-84 in page 2).
Point 6: Please add some information on expression level of TPP family members. This is relevant especially considering the strikingly low relative expression level (max. 0.0065) in different tissues of Jatropha
Response 6: We added relevant information on expression level of TPP family members in results section (Lines 112-116 in page 3).
Point 7: Results
P3 L 96- please explain more in detail while authors decided to work with JcTPP. This gene seems to be more universally expressed throughout plant tissues, while other members seem to be more flower and inflorescence specific.
Response 7: We added detailed information in results section (Lines 116-118 in page 3).
Point 8: Fig. 4 – please indicate how many plants of the T3 generation of each line were evaluated and whether differences were significant.
Response 8: We have added the detailed information on T3 generation plants in figure legend, materials and methods sections. Significance analysis was performed on relevant data in the revised manuscript (Lines 148-149 in page 5 and Lines 322-325 in page 11).
Point 9: The relatively low expression incrementation in L2 is not strongly reflected in phenotype – please comment on this.
Response 9: Thanks for your suggestions. The expression of JcTPPJ in transgenic line L2 seems to be lower than that in lines L1 and L4, however, which may be sufficient to induce phenotypic changes in transgenic Arabidopsis.
Point 10: Discussion:
P7 L90 … demonstrated that OsTPP7 activity sucrose availability – please correct sentence.
Response 10: We have corrected the sentence in discussion section (Line 222 in page 9).
Point 11: In general, authors should discuss more in detail the function that an altered sucrose metabolism could have on flowering time and flower development.
Response 11: We have added more discussions in both introduction and discussion sections (Lines 70-81 in page 2 and lines 224-230 in page 9).

Round 2
Reviewer 2 Report
The authors answered satisfactorily to all comments